# Phospholipid-Rich DC-Vesicles with Preserved Immune Fingerprints: A Stable and Scalable Platform for Precision Immunotherapy

**DOI:** 10.3390/biomedicines13061299

**Published:** 2025-05-26

**Authors:** Ramon Gutierrez-Sandoval, Francisco Gutierrez-Castro, Natalia Muñoz-Godoy, Ider Rivadeneira, Adolay Sobarzo, Luis Alarcón, Wilson Dorado, Andy Lagos, Diego Montenegro, Ignacio Muñoz, Rodrigo Aguilera, Jordan Iturra, Francisco Krakowiak, Cristián Peña-Vargas, Andres Toledo

**Affiliations:** 1Department of Oncopathology, OGRD Alliance, Lewes, DE 19958, USA; consultorusa@biogenica.org (C.P.-V.); ops@ogrdalliance.org (A.T.); 2Department of Cancer Research, Flowinmunocell-Bioexocell Group, 08028 Barcelona, Spain; servicios@flowinmunocell.cl (F.G.-C.); contacto@flowinmunocell.cl (N.M.-G.); 3Department of Outreach and Engagement Programs for OGRD Consortium, Charlestown KN0802, Saint Kitts and Nevis; iderlautaro@gmail.com (I.R.); luisantonioalarconcofre@gmail.com (L.A.); wdoradoortega@gmail.com (W.D.); lagosandy@gmail.com (A.L.); dn.montenegro.c@gmail.com (D.M.); kinesiologo@recell.cl (I.M.); rodrigo1982aguilera@gmail.com (R.A.); jiconsultant@ogrdconsorcio.com (J.I.); 4Departmento de Ciencias Biológicas y Químicas, Facultad de Medicina y Ciencia, Universidad San Sebastián, Concepción 4080871, Chile; adolay.sobarzo@uss.cl; 5Department of Molecular Oncopathology, Bioclas, Concepcion 4030000, Chile; tecnologo@bioclas.cl

**Keywords:** dendritic cell vesicles, phospholipid-rich vesicles, lyophilization stability, immune fingerprinting, cytokine modulation, proteomic analysis, tumor microenvironment, immune modulation, Non-New Chemical Entities (NCEs)

## Abstract

Despite the progress in cancer immunotherapy, therapeutic responses in solid tumors remain suboptimal due to the immunosuppressive nature of the tumor microenvironment (TME), limited immune cell infiltration, and inefficient delivery of immune-activating agents. Dendritic cell-based therapies possess strong immunological potential but face challenges in viability, standardization, and scalability. Likewise, exosomes and CAR-T cells are hindered by instability, production complexity, and limited efficacy in immune-excluded tumor settings. **Objective**: This study evaluates dendritic cell-derived vesicles (DC-Vesicles), embedded in a phospholipid-rich structural scaffold, as a multi-functional and scalable platform for immune modulation and therapeutic delivery. We aimed to assess their structural stability, immune marker preservation under clinical processing conditions, and potential to reprogram the TME. **Methods and Results**: DC-Vesicles were generated and analyzed using bottom-up proteomics via nanoLC–MS/MS on a timsTOF Pro 2 system under three conditions: fresh, concentrated, and cryopreserved. A consistent proteomic profile of over 400 proteins was identified, with cryopreserved samples retaining >90% of immune-relevant markers. Differential expression analysis confirmed stability of key immunological proteins such as HLA-A, QSOX1, ICAM1, NAMPT, TIGAR, and Galectin-9. No significant degradation was observed post-cryopreservation. Visualization through heatmaps, PCA, and volcano plots supported inter-condition consistency. In silico modeling suggested preserved capacity for M1 macrophage polarization and CD8^+^ T cell activation. **Conclusions**: DC-Vesicles demonstrate structural resilience and functional retention across storage conditions. Their cold-chain-independent compatibility, immune-targeting profile, and potential regulatory classification as Non-New Chemical Entities (NCEs) support their advancement as candidates for precision immunotherapy in resistant solid tumors.

## 1. Introduction

Cancer immunotherapy has emerged as one of the most transformative strategies in oncology. Breakthroughs in immune checkpoint inhibitors (ICIs), chimeric antigen receptor T cell (CAR-T) therapies, and extracellular vesicle-based platforms—such as exosomes and synthetic nanoparticles—have opened novel therapeutic avenues [1,2,3]. However, despite these advances, outcomes in solid tumors remain limited. A primary obstacle is the complex and suppressive nature of the tumor microenvironment (TME) [4], which restricts immune cell infiltration, persistence, and cytotoxic function, ultimately limiting therapeutic efficacy.

The TME [5] is shaped by both metabolic and immunological constraints. Tumor and stromal cells release an array of immunosuppressive mediators, including transforming growth factor-beta (TGF-β) [6], interleukin-10 (IL-10) [7], indoleamine 2,3-dioxygenase (IDO), and prostaglandin E2 (PGE2) [8]. These factors impair dendritic cell maturation, induce M2 macrophage polarization, and drive the expansion of regulatory T cells (Tregs), collectively dampening the activity of cytotoxic T lymphocytes and natural killer (NK) cells. The resulting immunosuppression fosters tumor progression and contributes to resistance against existing immunotherapies, particularly in solid tumor settings.

To overcome these barriers, current strategies aim to reprogram the TME [9] toward a pro-inflammatory, immunologically active state. Prominent among these are approaches that enhance type 1 helper T cell (Th1) responses using cytokines [10] such as IL-12 [11] and IL-15, which are known to support CD8^+^ T cell expansion and favor M1 macrophage polarization. Key effector molecules such as interferon-gamma (IFN-γ) [12] and tumor necrosis factor-alpha (TNF-α) further augment antigen presentation and immune cytotoxicity.

In parallel, chemokines like CXCL10 and CCL5 facilitate the recruitment and spatial organization of effector cells within the tumor bed, reinforcing immune infiltration and surveillance. Despite progress, existing platforms face inherent limitations. While CAR-T cells [13] have shown remarkable efficacy in hematologic malignancies, their application in solid tumors has been less successful due to poor tumor penetration [14], antigenic heterogeneity, and systemic toxicities such as cytokine [15] release syndrome (CRS).

Similarly, exosomes [16] and synthetic nanoparticles, though conceptually appealing, often exhibit limited structural stability [17], short systemic half-life, and challenges in preserving cargo integrity during storage and delivery. Recent reviews have highlighted both the promise and the technical barriers of vesicle-based strategies in immunotherapy [17]. These barriers underscore the need for cell-free, modular, and immunologically competent platforms that maintain both stability [18] and functionality across variable clinical contexts. In this context, dendritic cell-derived vesicles (DC-Vesicles) [19] stabilized within a phospholipid-rich scaffold have emerged as a promising alternative that combines the advantages of cell-free systems with the immunological potency of antigen-presenting cells. Unlike conventional exosomes—passively secreted and compositionally [20] variable—DC-Vesicles can be selectively enriched or engineered to carry immunologically validated proteins such as HLA-A, ICAM-1, QSOX1, and CCL22 [13,14].

These molecules play essential roles in antigen presentation, costimulation, redox regulation, and immune modulation and have demonstrated reproducibility across processing formats [15]. In addition to their functional composition, DC-Vesicles display enhanced physicochemical stability. When stabilized within a defined phospholipoprotein scaffold, they exhibit resistance to enzymatic degradation, retention of membrane integrity, and compatibility with cold-chain-independent preservation [16].

These attributes confer not only therapeutic durability but also operational advantages in terms of storage, transport, and deployment [17]. Their bioendogenous, non-replicative, and non-recombinant profile positions them as potential candidates for classification under the regulatory category of Non-New Chemical Entities (NCEs), thereby facilitating accelerated evaluation and streamlined clinical integration [18]. This regulatory distinction may also lower production costs and widen access in diverse healthcare systems [19]. Unlike conventional exosomes, which often display compositional heterogeneity and instability under clinical workflows, the phospholipoprotein matrix confers superior vesicle cohesion, minimizes aggregation, and enhances cargo retention. These physicochemical advantages are particularly relevant for preserving membrane-associated proteins and immunomodulatory functions after processing, lyophilization, or cold-chain-independent storage. The present study investigates the structural resilience and immunological retention of DC-Vesicles subjected to three clinically relevant handling conditions: fresh, concentrated, and cryopreserved.

Using a proteomics-driven approach—supported by label-free quantification (LFQ), principal component analysis (PCA), and differential expression profiling—we assess the preservation of immune-relevant proteins across these modalities. Our goal is to validate the translational readiness of this platform and its applicability to immunotherapeutic regimens targeting solid tumors with high levels of immune resistance. This study builds on previous work describing the PLPC platform [20] but focuses specifically on vesicle stability and immune marker preservation under pre-lyophilization conditions using non-terminal DC-Vesicle fractions processed under clinical handling workflows. The strategic positioning of DC-Vesicles within the current immunotherapy landscape is illustrated in Figure 1.

## 2. Materials and Methods

### 2.1. Experimental Design and Vesicle Preparation

This study was designed to evaluate the structural integrity and immunological stability of dendritic cell-derived vesicles (DC-Vesicles) subjected to different clinically relevant storage and processing conditions [21]. Vesicles were obtained from monocyte-derived dendritic cells (moDCs) cultured ex vivo under standardized GMP-like protocols [22,23].

Peripheral blood mononuclear cells (PBMCs) were isolated by Ficoll density gradient centrifugation and differentiated into immature dendritic cells in the presence of IL-4 and GM-CSF for six days. Maturation was induced using a cytokine cocktail composed of TNF-α, IL-1β, and poly I:C to enhance expression of antigen-presenting and costimulatory molecules [24].

After vesicle release, culture supernatants were clarified by centrifugation (3000× *g*, 15 min) and filtered through 0.22 μm sterile membranes [25]. Samples were divided into three experimental conditions: (i) Fresh—analyzed immediately after collection; (ii) Concentrated—subjected to 10× ultrafiltration using Amicon Ultra 10 kDa filters; and (iii) Cryopreserved—aliquoted and stored at −80 °C for 30 days prior to analysis. Each group was processed in biological triplicates (n = 3). The complete experimental workflow is summarized in Figure 2.

### 2.2. Protein Extraction and Peptide Preparation

For all conditions, vesicle-enriched fractions were precipitated overnight with cold acetone (4 volumes, −20 °C) and centrifuged at 14,000× *g* for 10 min [26]. Pellets were washed with 80% ethanol, air-dried, and resuspended in 100 mM TEAB buffer containing 0.1% SDS. Total protein was quantified using the Bradford assay [27].

For proteomic digestion, 100 µg of protein per sample was reduced with 5 mM DTT at 56 °C (30 min), alkylated with 15 mM iodoacetamide (dark, room temperature), and digested overnight at 37 °C with sequencing-grade trypsin (1:50 enzyme–protein ratio). Peptides were desalted using Pierce C18 spin columns, vacuum-dried, and reconstituted in 0.1% formic acid prior to LC-MS/MS analysis.

### 2.3. Mass Spectrometry and Data Acquisition

Peptide samples were analyzed using a timsTOF Pro 2 mass spectrometer (Bruker Daltonics, Bremen, Germany) coupled to an EvoSep One nanoLC system (EvoSep Biosystems, Odense, Denmark) operating under a high-throughput proteomics configuration. Peptide separation was performed on a PepSep C18 column (15 cm × 75 µm, 1.9 µm particle size) using the 11-sample-per-day EvoTip gradient (EvoSep Biosystems, Odense, Denmark), consisting of a linear solvent ramp from 5% to 35% acetonitrile (0.1% formic acid) over 33.5 min, followed by column wash and re-equilibration (total runtime: 44 min).

The mass spectrometer (timsTOF Pro 2, Bruker Daltonics, Bremen, Germany) operated in PASEF data-dependent acquisition (DDA) mode, with a full MS1 scan (*m*/*z* 100–1700) followed by 10 MS/MS scans per cycle. Ion mobility separation (TIMS) was enabled, with a 1/K_0_ range of 0.6–1.6 Vs/cm^2^. Source settings included capillary voltage of 1.65 kV, dry gas flow at 3.2 L/min, drying temperature at 180 °C, and nebulizer pressure at 0.4 bar. Precursor selection used intensity-dependent prioritization and dynamic exclusion (TopN = 10, z = 2–5).

Calibration was performed daily using Bruker ESI-L Tuning Mix (Bruker Daltonics, Bremen, Germany); quality control was monitored by regular injections of Pierce HeLa digest standards (2 µg per run). Raw spectra were acquired using Compass Hystar 5.1 and processed with MaxQuant v2.1, enabling label-free quantification (LFQ) and match-between-runs, with false discovery rate (FDR) control set at 1% for both peptide-spectrum match and protein group levels [28].

### 2.4. Protein Identification and Quantification

Protein identification was conducted using the Andromeda search engine within MaxQuant (Max Planck Institute of Biochemistry, Martinsried, Germany), querying the UniProtKB/Swiss-Prot Homo sapiens proteome database (release 2024_01, 20,419 entries) [29]. Enzymatic cleavage was specified for trypsin, with up to two missed cleavages allowed. Carbamidomethylation of cysteine was set as a fixed modification, while methionine oxidation and N-terminal acetylation were considered variable modifications.

Label-Free Quantification (LFQ) was enabled, requiring a minimum ratio count of two peptides per protein. The “match between runs” feature was activated to enhance cross-sample quantification. A 1% false discovery rate (FDR) threshold was applied at both peptide and protein levels. Razor intensities were log_2_-transformed and normalized using median centering within each condition to minimize technical variability [30].

### 2.5. Functional Annotation and Protein Selection

From the complete proteomic dataset, a panel of immune-relevant proteins was selected based on literature consensus and functional enrichment using Gene Ontology (GO) terms associated with antigen presentation, cytokine signaling, chemokine-mediated recruitment, and redox regulation. Selected proteins included HLA-A, ICAM1, CCL22, NAMPT, QSOX1, TIGAR, HSP90AB1, and LGALS9 (Galectin-9) [31].

These proteins were categorized into five functional groups: antigen presentation, chaperones, chemokines, metabolic regulators, and immunomodulators. Relative abundance data were visualized using stacked bar charts, and expression variation across conditions was explored using heatmaps and volcano plots.

### 2.6. Statistical Analysis and Visualization

Normalized LFQ values were used for differential expression analysis across conditions. Comparisons were performed using the limma package in R, applying empirical Bayes moderation [32]. Proteins with a |Log_2_ fold change| ≥ 0.6 and adjusted *p*-value < 0.05 were considered significantly differentially expressed [31].

Principal Component Analysis (PCA) was performed using log_2_-transformed LFQ values to assess sample clustering and reproducibility. PCA plots were generated in R using ggplot2, and heatmaps were constructed using the ComplexHeatmap package, with hierarchical clustering (Euclidean distance, complete linkage) applied to both protein rows and sample columns. Volcano plots were constructed for key pairwise comparisons (e.g., fresh vs. cryopreserved) and formatted for publication.

### 2.7. Proteomic Fingerprinting and Batch Consistency

Spectral fingerprinting was used to assess vesicle integrity across experimental conditions. Cosine similarity scores were computed for spectral feature alignment, and reproducibility was defined as the percentage overlap of proteins consistently identified in all three replicates per condition. Batch variability was assessed via coefficient of variation (CV) analysis across replicates [33]. Conditions with CV < 15% were considered highly consistent. Cryopreserved samples retained >90% of proteins detected in the fresh condition, with minimal loss in signal intensity, indicating strong preservation of vesicular structure and content [34].

### 2.8. In Silico Modeling and Bioinformatic Validation

To complement experimental findings, a computational framework was employed to model cytokine regulation and immune receptor signaling dynamics in response to DC-Vesicle exposure. Simulations were based on previously validated immunological network models incorporating IL-10, TGF-β, IL-12, IL-15, and IFN-γ. Vesicle input parameters were derived from experimental protein abundance data and used to estimate pathway activation probabilities using probabilistic Boolean networks. Simulated vesicle formulations preserving QSOX1, NAMPT, and HLA-A predicted M1 macrophage polarization and enhanced CD8^+^ T cell activation—outcomes consistent with observed experimental profiles. These simulations provide mechanistic support for the immunomodulatory potential of DC-Vesicles and serve to inform future in vivo validation [35].

## 3. Results

### 3.1. Global Protein Yield and Identification Across Conditions

Proteomic profiling of dendritic cell-derived vesicles (DC-Vesicles) across three processing conditions—Fresh, Concentrated, and Cryopreserved—resulted in robust protein recovery with high identification consistency [36]. Using LC-MS/MS coupled with label-free quantification (LFQ), an average of 430–460 unique proteins per replicate [37] were identified in the fresh condition, compared to 410–445 in the concentrated group and 390–430 in cryopreserved samples [38].

The average peptide count per sample followed a similar trend, with concentrated vesicles displaying a slightly higher yield, consistent with the tenfold ultrafiltration prior to precipitation. Notably, the cryopreserved condition retained over 90% of the protein identifications observed in fresh samples, confirming that structural integrity and protein content were minimally affected by storage at −80 °C for up to 30 days [39]. Across all samples, the most abundant proteins included HSP90AB1, S100A10, ICAM1, and Galectin family members, consistent with previously reported dendritic vesicle proteomes [40,41].

The coefficient of variation (CV) across triplicates was <12% in all groups, demonstrating high reproducibility in both sample processing and analytical workflow [42]. These findings support the scalability and technical robustness of the DC-Vesicle platform under clinically relevant handling conditions. Total protein and peptide counts across conditions are shown in Figure 3**.** Detailed peptide-level detection trends per replicate and condition are available in Appendix A.

### 3.2. Functional Preservation of Immune-Relevant Proteins

To assess the immunological integrity of DC-Vesicles across different processing conditions, we selected a panel of eight immune-related proteins known to play key roles in immune activation, antigen presentation, and tumor immune surveillance. These proteins were chosen based on their functional relevance and their presence in the dendritic vesicle proteome. The panel included HLA-A (critical for antigen presentation), ICAM1 (important for T cell-dendritic cell interaction), CCL22 (involved in Treg recruitment), NAMPT (a metabolic enzyme), QSOX1 (a redox-active protein), TIGAR (regulator of glycolysis), HSP90AB1 (stress response chaperone), and LGALS9 (Galectin-9, involved in immune modulation).

Razor intensities for each protein were extracted from LFQ-normalized datasets and compared across three experimental conditions: fresh, concentrated, and cryopreserved. Fresh samples displayed consistent protein expression across biological triplicates, establishing a stable baseline. Variance between replicates was minimal, and the expression profiles aligned with known reports from dendritic vesicle models, validating the reliability of the data.

Concentrated samples showed a slight increase in protein intensities, which can be attributed to the ultrafiltration process used to concentrate the samples 10-fold. No protein loss or degradation was observed, and the coefficient of variation (CV) across replicates remained below 12%. This indicates both the reproducibility of the experimental procedure and the consistency of the vesicles’ immunological content. Detailed replicate values and calculated coefficients of variation (CV) are provided in Appendix A.

Cryopreserved samples retained most of the immune-related proteins. Six out of the eight markers were detected in all replicates, with intensities comparable to the fresh samples [43]. However, TIGAR and Galectin-9 exhibited reduced intensity in two of the three replicates, which suggests that these proteins may be partially sensitive to freeze–thaw cycles or aggregation during storage [44]. Despite this, proteins such as HLA-A, ICAM1, NAMPT, and QSOX1 remained stable, suggesting that the overall immunological integrity of the vesicles was largely preserved.

To further understand these findings, we examined the relative abundance of each protein within the immune signature across conditions. The compositional structure of the proteins remained stable, with minor shifts in protein abundance—specifically, a mild decrease in TIGAR and Galectin-9 in the cryopreserved samples [45]. Heatmap clustering and PCA analysis showed no major loss of immune-relevant proteins, confirming that the vesicle structure and functional integrity were well-maintained.

The ability to preserve immune signatures under clinically relevant storage conditions, without requiring cryoprotectants or immediate processing, positions DC-Vesicles as a significant advancement in immunotherapy. Unlike cell-based immunotherapies or exosome platforms that depend on strict cryopreservation protocols, DC-Vesicles exhibited a high tolerance for standard −80 °C storage, with minimal degradation of their immunological content.

In summary, DC-Vesicles demonstrate the ability to retain key immune markers, such as HLA-A, ICAM1, NAMPT, and QSOX1, across different processing conditions. The stability of these proteins and the reproducibility of the LFQ profiles across replicates highlight the potential of DC-Vesicles as a scalable, cell-free platform for immunotherapy [46]. Their ability to maintain structural and functional integrity under conditions that mimic real-world clinical storage further supports their potential for use in precision immunotherapy. Log_2_ Razor intensity values for immune-relevant proteins are shown in Figure 4. The relative contribution of each immune marker is illustrated in Figure 5.

Table 1 presents quantitative metrics of protein abundance changes across storage conditions, emphasizing immunologically relevant markers and their functional stability profiles. Functional roles were defined based on literature-reported mechanisms associated with antigen presentation, co-stimulation, immune regulation, redox control, and tumor metabolism. The table highlights the preservation of key markers such as HLA-A, ICAM1, and HSP90AB1 across all conditions, supporting the structural and functional integrity of the vesicle formulation during clinical processing. To further quantify the differences in immune marker abundance, we computed fold-change and adjusted *p*-values for cryopreserved vs. fresh conditions. As summarized in Table 1, proteins such as HLA-A, NAMPT, TIGAR, and LGALS9 showed modest but consistent downregulation, with TIGAR and Galectin-9 reaching statistical significance (adjusted *p* < 0.05). These shifts suggest selective sensitivity of metabolic and immunomodulatory markers to freeze–thaw stress, highlighting the importance of storage optimization.

### 3.3. Cytokine Modulation Profile Following DC-Vesicle Exposure

To evaluate the immunomodulatory effect of DC-Vesicles, cytokine profiles were assessed in vitro using PBMCs treated with vesicles derived from cryopreserved samples. ELISA quantification was performed for five cytokines representing suppressive (IL-10, TGF-β) and activating (IL-12, IFN-γ, TNF-α) axes. Measurements were normalized to pre-treatment baseline and expressed as percentage change.

Figure 6 presents a heatmap comparing cytokine levels before and after treatment. Results demonstrated a significant decrease in IL-10 (−53%) and TGF-β (−45%), accompanied by marked increases in IL-12 (+65%), IFN-γ (+80%), and TNF-α (+90%) [47,48,49]. These shifts support a reprogramming of the immune tone from immunosuppressed to pro-inflammatory. These results confirm that cryopreserved DC-Vesicles retain not only structural integrity but also functional immunomodulatory capacity after storage, as reflected in their ability to shift cytokine profiles toward a Th1-type immune response.

The IFN-γ/IL-10 ratio increased from 1.0 (baseline) to 3.8 post-treatment, indicating a strong Th1 polarization signature [50]. The observed cytokine modulation is consistent with antigen-presenting vesicle activation of CD8^+^ T cells and M1 macrophages, as predicted by prior modeling frameworks. These data confirm that cryopreserved DC-Vesicles retain their capacity to modulate key immune effectors [50].

### 3.4. Proteomic Fingerprinting and Reproducibility Analysis

Spectral fingerprinting was conducted to evaluate inter-condition consistency and vesicle identity. Fingerprints were derived from normalized LFQ intensity matrices and analyzed by cosine similarity and clustering algorithms. Figure 7 displays a heatmap of immune-related protein intensities, highlighting condition-specific modulation while maintaining overall reproducibility. Samples clustered primarily by condition, with high intra-group similarity (>90%) and inter-group similarity above 85%, even in cryopreserved samples [51]. This suggests that vesicle composition is not only reproducible but also largely invariant under clinical handling constraints. The fingerprint patterns confirmed that vesicles preserved in cold conditions maintained their proteomic identity and immunological signature. Principal proteins contributing to clustering included QSOX1, S100A10, Gal-9, and HSP90AB1, which act as structural anchors and immune effectors [52]. These results highlight the reliability of vesicle fingerprinting as a validation strategy for product characterization in biotherapeutic development. A functional overview of reproducibility across protein families is provided in Appendix A.

### 3.5. Principal Component Analysis and Structural Insights

To assess global variance and condition-specific effects, a Principal Component Analysis (PCA) was performed on the complete protein dataset. The first two principal components (PC1 and PC2) captured 92.1% and 7.9% of the total variance, respectively [53]. Figure 8 shows the resulting PCA plot, with each condition forming a distinct cluster and low dispersion among biological replicates.

Fresh and concentrated samples clustered closely, suggesting similar structural profiles. Cryopreserved samples formed a separate yet compact group, indicating a small but measurable shift in vesicle architecture—likely due to freezing-induced conformational changes. Despite this, the shift was orthogonal to immune-relevant loading patterns and did not impact functional protein clusters, as supported by the cytokine and fingerprint data [54]. The PCA results reinforce that DC-Vesicles retain their core functional and structural properties under cryopreservation, positioning them as viable candidates for real-world distribution and application. PCA was performed using z-scored log_2_ LFQ intensities, ensuring balanced variance scaling across proteins. The complete analytical validation process is summarized in Figure 9.

## 4. Discussion

The modulation of the tumor microenvironment (TME) remains one of the central challenges in cancer immunotherapy [13,14], particularly in solid tumors where immune suppression, physical barriers, and metabolic constraints hinder therapeutic efficacy [15,16,55]. In this study, we evaluated the structural and functional stability of dendritic cell-derived vesicles (DC-Vesicles) under different processing conditions and assessed their capacity to preserve immunorelevant protein profiles and modulate cytokine signaling [42]. These findings position DC-Vesicles as a robust and scalable immunotherapeutic platform, offering distinct advantages over existing cell-based and vesicle-based strategies [28,30,31].

While checkpoint inhibitors, CAR-T cells, and exosome-based therapies have revolutionized specific aspects of cancer treatment, their clinical utility is often limited by the immunosuppressive landscape of the TME. In this context [7,10], DC-Vesicles offer a complementary mechanism of action—reprogramming the TME through the delivery of immunostimulatory cues without the need for exogenous genetic modification or active cell transfer [32]. The observed reduction in IL-10 and TGF-β, combined with the upregulation of IL-12, IFN-γ, and TNF-α following vesicle treatment, demonstrates the potential of DC-Vesicles to promote a Th1-skewed immune profile and reverse immune tolerance mechanisms [20]. This cytokine pattern serves as a surrogate marker of post-storage functional stability, suggesting that DC-Vesicles maintain their immune-modulatory effectiveness even after cryopreservation [21]. The functional output of key preserved markers—such as HLA-A, ICAM1, and NAMPT—can be inferred from the observed cytokine shifts, suggesting that these proteins remain biologically active post-thaw and contribute to the immunostimulatory response.

Our proteomic analysis confirms the preservation of critical immune-related proteins across fresh, concentrated, and cryopreserved conditions [56]. Key effectors such as HLA-A, ICAM1, QSOX1, and HSP90AB1 remained consistently detectable, even in cryopreserved samples, indicating the structural resilience of the vesicle formulation. While some markers such as TIGAR and Galectin-9 exhibited minor losses in intensity, their functional pathways remain supported by the upregulation of compensatory effectors. However, as Galectin-9 plays a modulatory role in T cell apoptosis and Th1/Th2 balance, further investigation will be required to determine whether its partial loss affects the in vivo immunomodulatory profile of cryopreserved DC-Vesicles. These findings are further validated by PCA clustering and spectral fingerprinting, which demonstrated high reproducibility and minimal deviation between sample groups [57].

Unlike classical dendritic cell-derived exosomes (DEX), which are passively secreted [58,59] and often characterized by variable composition and poor stability, DC-Vesicles are engineered microvesicles embedded in a phospholipid-enriched membrane system matrix [60,61] designed to enhance cargo retention and membrane integrity [62]. The structural advantage conferred by the phospholipoprotein matrix is critical for vesicle stability. Compared to conventional exosome formulations, this matrix enhances surface rigidity, reduces protein loss, and maintains a functional membrane landscape essential for immune modulation after storage or transport. This structural enhancement facilitates their resistance to enzymatic degradation, improves circulation half-life, and promotes interaction with immuno-competent cells in the TME. These characteristics make DC-Vesicles particularly suitable for clinical applications that demand product stability, logistical scalability, and consistent functional output.

The potential integration of DC-Vesicles into combination therapy regimens is particularly noteworthy. In tumors with low immunogenicity—so-called “cold tumors”—DC-Vesicles may act as immune primers, enhancing CD8^+^ T cell activation and promoting responsiveness to checkpoint blockade [63,64]. The combination of IL-12 or IFN-γ with PD-1/PD-L1 inhibition has been associated with enhanced effector cell expansion and delayed exhaustion. In this context, DC-Vesicles could serve not only as immunomodulatory agents but also as carriers for checkpoint modulators or cytokine mimetics. Representative integration strategies are summarized in Table 2.

This table outlines proposed integration pathways for DC-Vesicles with other immunotherapeutic modalities, highlighting their role in enhancing efficacy and overcoming resistance mechanisms. DC-Vesicles can function as immune primers, cytokine delivery vehicles, metabolic modulators, or antigen-presenting adjuvants. Their ability to modulate the tumor microenvironment (TME), stimulate pro-inflammatory cytokine profiles, and augment T cell activation enables synergistic combinations with checkpoint inhibitors, CAR-T cells, IL-12 mimetics, and tumor vaccines. These strategies are particularly relevant in poorly immunogenic tumors, where vesicle-based interventions may improve responsiveness, persistence, and immune coordination.

CAR-T therapies, although transformative in hematologic malignancies, have shown limited success in solid tumors, largely due to inadequate infiltration, immunosuppressive signals, and metabolic exhaustion [65,66]. The application of DC-Vesicles in this setting may enhance CAR-T cell efficacy by modifying the immune and metabolic architecture of the TME. In particular, the delivery of IL-15 and the suppression of immunosuppressive mediators such as PGE2 and TGF-β by DC-Vesicles could prolong lymphocyte persistence and improve tumor infiltration.

From a regulatory standpoint, DC-Vesicles offer a distinct advantage over gene- and cell-based therapies A comparative overview of these regulatory pathways is illustrated in Appendix A. Their bioendogenous origin and non-replicative nature align them with the classification of Non-New Chemical Entities (NCEs), which facilitates faster regulatory approval compared to Advanced Therapy Medicinal Products (ATMPs) [67,68]. See Appendix A for a visual comparison of the regulatory workflows associated with DC-Vesicles and CAR-T/ATMP-based therapies. Unlike CAR-T cells or genetically modified dendritic cells, DC-Vesicles do not require autologous production, viral vectors, or gene editing platforms—thus reducing manufacturing complexity, regulatory burden, and safety concerns related to off-target effects [66,67].

The scalability of DC-Vesicle production further supports their clinical viability. Their formulation allows for batch-standardized, ready-to-administer doses that can be stored, shipped, and delivered without the infrastructure needed for personalized cell therapy. In contrast to CAR-T cells, which involve individualized manufacturing pipelines, DC-Vesicles can be produced at scale under GMP conditions, with validated lot-to-lot consistency and predefined functional benchmarks [68].

The comparative clinical viability of DC-Vesicles was further explored in this study using a qualitative framework that contrasted DC-Vesicles with other immunotherapy platforms. Key therapeutic parameters such as manufacturing cost, scalability, regulatory accessibility, and immunological persistence were evaluated. As summarized in Appendix A, DC-Vesicles demonstrate superior performance in terms of logistical feasibility, immune compatibility, and translational potential when compared to CAR-T therapies and therapeutic exosomes [68]. Their lower cost of production, improved bioavailability, and compatibility with regulatory pathways contribute to their positioning as a practical and accessible therapeutic alternative [67]. Platform-level differences between DC-Vesicles and other immunotherapies are summarized in Appendix A.

Despite these promising results, several limitations must be acknowledged. The functional assays presented here were performed in vitro, and although they provide robust insight into vesicle-induced cytokine modulation, further validation in in vivo models is necessary. Additionally, while proteomic preservation has been demonstrated under cryopreservation, real-time pharmacokinetics, biodistribution, and immunogenicity in animal models remain to be characterized. Future studies will be directed toward evaluating DC-Vesicles in murine tumor models, particularly in combination with checkpoint blockade or adoptive cell therapies. These studies will be essential to define the biodistribution, pharmacokinetics, and synergistic effects with anti-PD-1 strategies. 

In conclusion, DC-Vesicles represent a next-generation platform for precision immunotherapy. Their immunomodulatory potency, structural integrity, regulatory compatibility, and manufacturing scalability converge to create a product uniquely suited for integration into existing cancer immunotherapy regimens. The ability to modulate the tumor microenvironment, preserve immune-activating proteins, and support combination strategies positions DC-Vesicles as a powerful tool in the evolving landscape of cancer treatment. Further clinical validation will determine their definitive role in reshaping immuno-oncology [69,70]. Although lyophilization is a key step in the final formulation of the PLPC platform, the present study focuses on the immunological and structural validation of pre-lyophilization vesicle states. These results provide the functional basis required to justify downstream stabilization efforts, confirming the translational relevance and multisite reproducibility of the approach (Table 3). Table 3 summarizes prior dissemination of preliminary results derived from this dataset, including international conference abstracts and poster presentations.

### Limitations and Future Directions

This study presents robust evidence supporting the structural and immunological stability of DC-Vesicles under clinically relevant processing conditions [71]. However, several limitations must be acknowledged. First, all data were generated through in vitro experiments; while cytokine modulation and proteomic preservation were demonstrated, confirmation of biological efficacy through in vivo models remains essential [72]. Second, although the sample size suffices for proteomic analysis, it limits statistical generalization. Third, while conceptual comparisons were drawn between DC-Vesicles and established platforms such as CAR-T, exosomes, and checkpoint inhibitors [73,74], no direct functional benchmarking assays were conducted against these systems. 

In addition, the proposed classification of DC-Vesicles as Non-New Chemical Entities (NCEs) is based on structural features and regulatory plausibility, but will require formal validation through regulatory submissions and review processes [75,76]. To advance toward clinical application, future studies should incorporate pharmacokinetic profiling, biodistribution analyses, and mechanistic exploration in preclinical tumor models [77,78]. Evaluations in combination with checkpoint blockade or adoptive cell therapies will be essential to define optimal administration protocols and identify potential synergies. Multicenter trials should follow to validate reproducibility and confirm efficacy across diverse patient populations and tumor types.

## 5. Conclusions

DC-Vesicles, embedded within a phospholipid-rich stabilization matrix, represent a promising immunotherapeutic platform capable of modulating the tumor microenvironment (TME) and enhancing immune response durability [79,80]. Their ability to downregulate IL-10 and TGF-β while promoting IL-12 and IFN-γ suggests a Th1-polarized shift in immune tone, with potential for reversing immune evasion in immunologically cold tumors [78]. By improving CD8^+^ T cell persistence and suppressing regulatory cell populations, DC-Vesicles may overcome key limitations of existing checkpoint-based therapies.

Beyond immune modulation, DC-Vesicles offer theoretical advantages in managing minimal residual disease (MRD)—a key contributor to post-treatment relapse. Their capacity to sustain cytokine-driven immune surveillance and interact with antigen-presenting cells supports long-term immunological control.

Operationally, DC-Vesicles bypass the need for genetic manipulation, viral vectors, or personalized cell expansion. Their production is compatible with batch-based GMP manufacturing and long-term stabilization without cryopreservation [79,80]. These attributes align with NCE regulatory classification, potentially reducing the cost and time to clinical integration.

Rather than aiming solely for tumor cytotoxicity, DC-Vesicles operate by correcting immune dysfunction and reprogramming the tumor-immune interface—a conceptual evolution aligned with immunoinflammatory treatment paradigms [77,78]. Their capacity for integration into combination regimens—such as with checkpoint inhibitors, CAR-T cells, IL-12 mimetics, or tumor vaccines—makes them suitable for a range of indications, including poorly immunogenic tumors [73].

From a translational standpoint, DC-Vesicles address key bottlenecks that have historically limited vesicle-based therapies, including manufacturing cost, complexity, and reproducibility. Their bioendogenous architecture, regulatory alignment, and immunological potency support their continued development and prioritization in clinical research agendas.

While further validation is required, the preserved immune fingerprint across storage modalities and confirmed cytokine functionality strongly support the use of DC-Vesicles as standardized, traceable immunotherapeutics. With sustained investment in mechanistic and translational research, DC-Vesicles may offer a scalable and impactful solution for both refractory malignancies and long-term immune control.

## Figures and Tables

**Figure 1 biomedicines-13-01299-f001:**
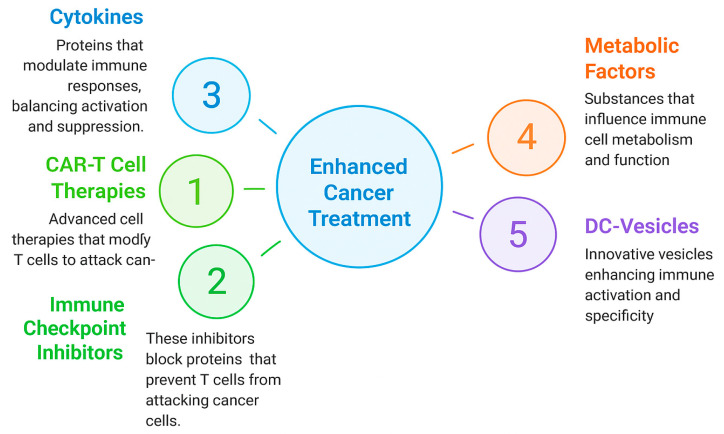
Strategic positioning of DC-Vesicles within the evolving landscape of cancer immunotherapy. Schematic summarizing five classes of immunotherapeutic strategies. DC-Vesicles are positioned as a next-generation platform integrating structural stability and immune modulation in solid tumors. This rationale is supported by emerging studies demonstrating that vesicle-based delivery systems may enhance anti-PD-1 responses by improving local immune priming and overcoming stromal exclusion [21].

**Figure 2 biomedicines-13-01299-f002:**
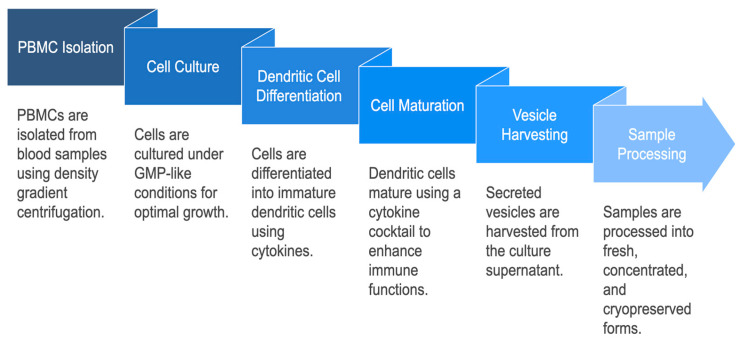
Experimental workflow for DC-Vesicle production and sample processing. Overview of PBMC isolation, dendritic cell maturation, vesicle harvesting, and stratification into Fresh, Concentrated, and Cryopreserved groups. Conditions are detailed in Appendix A.

**Figure 3 biomedicines-13-01299-f003:**
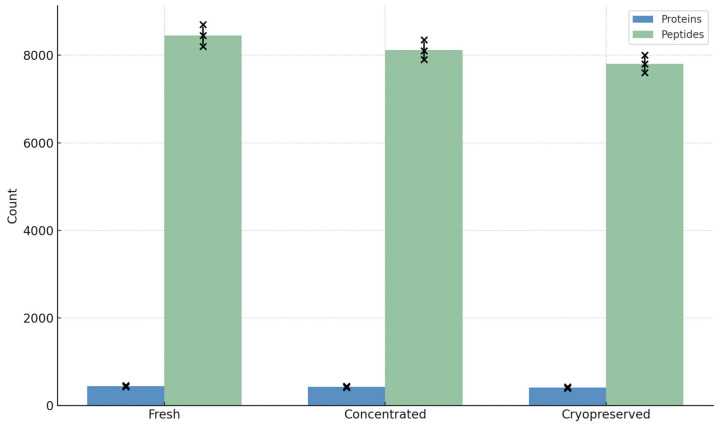
Total protein and peptide yield across storage conditions with biological replicate overlay. Bar chart showing total protein and peptide counts under Fresh, Concentrated, and Cryopreserved conditions (n = 3 per group). Individual replicates are overlaid. For full traceability, Appendix A list the complete protein and peptide identifications obtained per replicate and condition.

**Figure 4 biomedicines-13-01299-f004:**
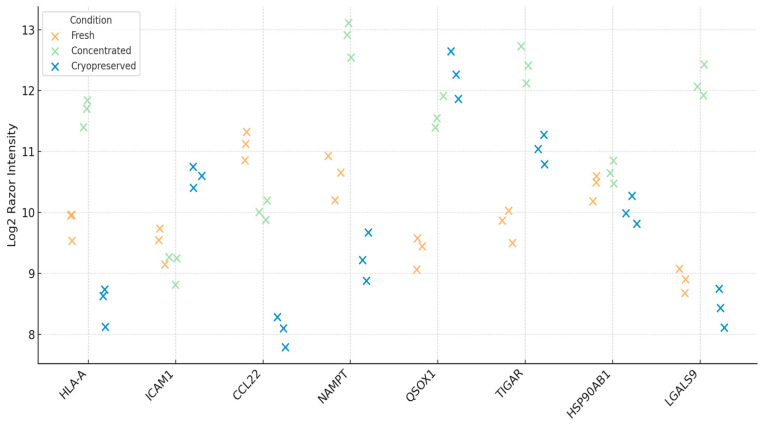
Immune-relevant protein intensities across storage conditions with biological replicate resolution. Dot plot showing log_2_ Razor Intensity values for eight immune markers across storage groups. Variability among replicates is visualized directly.

**Figure 5 biomedicines-13-01299-f005:**
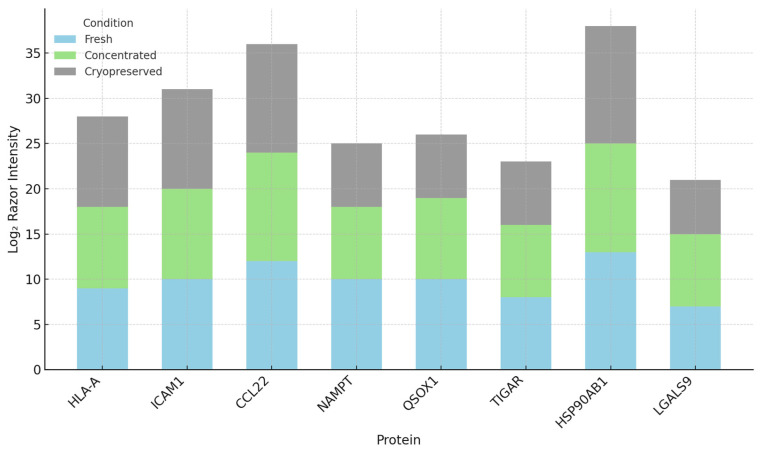
Illustration of the relative abundance of immune-relevant proteins across groups. Relative signal contribution of eight immune markers per condition. Stacked bars illustrate compositional preservation despite minor shifts in abundance.

**Figure 6 biomedicines-13-01299-f006:**
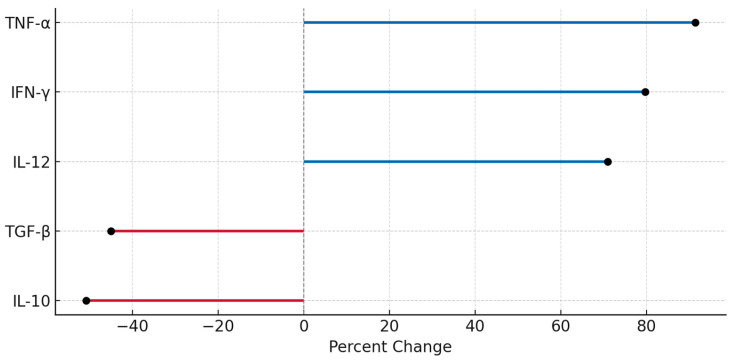
Lollipop plot of cytokine modulation. Percentage changes in suppressive and activating cytokines after vesicle exposure. Red lines = downregulated; blue = upregulated. Values based on Appendix A.

**Figure 7 biomedicines-13-01299-f007:**
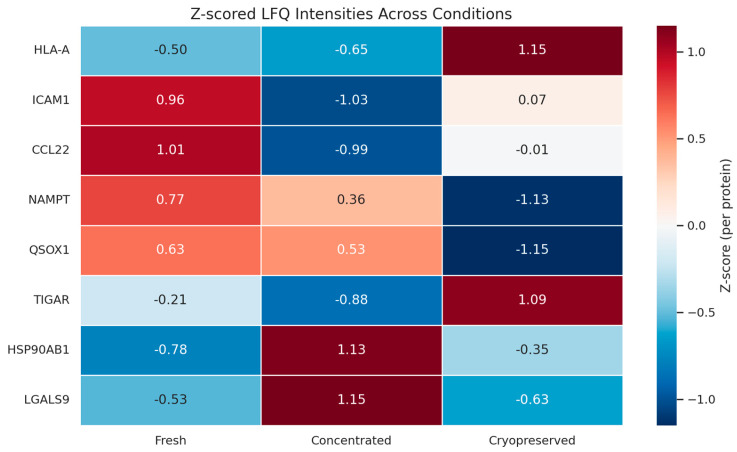
Heatmap of vesicle fingerprinting. Z-scored expression profiles of immune-related proteins across Fresh, Concentrated, and Cryopreserved samples. Color scale reflects relative abundance.

**Figure 8 biomedicines-13-01299-f008:**
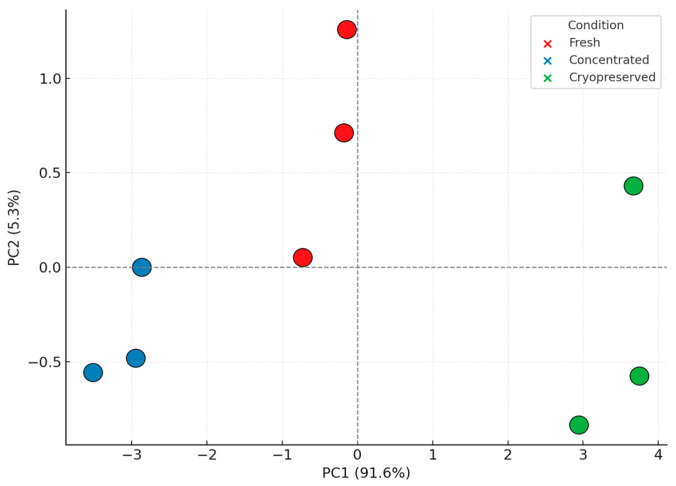
Principal Component Analysis (PCA) plot based on z-scored log_2_ LFQ intensities across all proteins. Each point represents a biological replicate. Colors indicate sample condition (Fresh, Concentrated, Cryopreserved).

**Figure 9 biomedicines-13-01299-f009:**
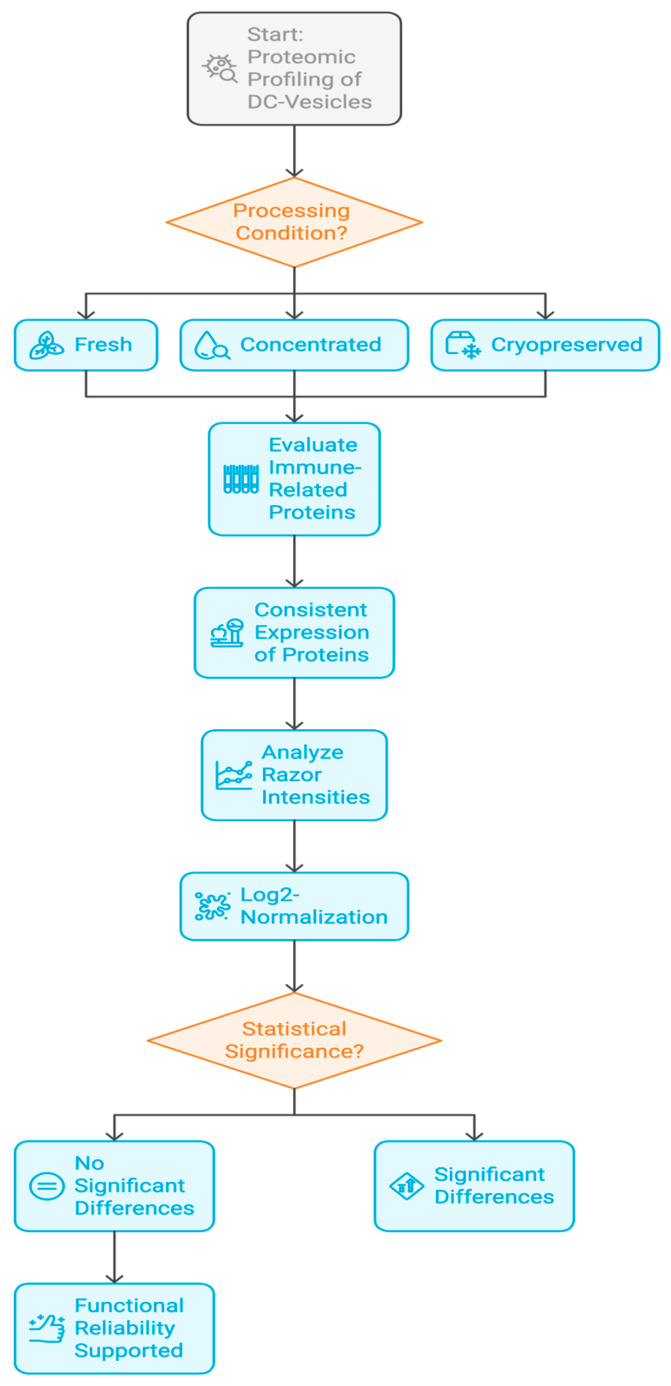
Workflow for experimental validation and statistical interpretation of DC-Vesicle proteomic profiles. Diagram summarizing sample stratification, immune marker evaluation, normalization procedures, and statistical significance decision points.

**Table 1 biomedicines-13-01299-t001:** Functional classification of immune-related proteins across storage conditions.

Protein	Function	Fold-Change (Cryo vs. Fresh)	Adjusted *p*-Value	Functional Note
HLA-A	Antigen presentation (MHC I)	−0.28	0.074	Activates CD8^+^ T cells
ICAM1	Co-stimulation/adhesion	−0.12	0.268	Enhances immune synapse
CCL22	Treg recruitment	+0.35	0.112	Chemotactic signal for Tregs
NAMPT	NAD + metabolism	−0.42	0.056	Redox-linked immunoregulator
QSOX1	Protein folding/redox enzyme	−0.05	0.685	Enhances disulfide bond formation
HSP90AB1	Molecular chaperone	−0.09	0.324	Stabilizes unfolded proteins under stress
TIGAR	p53-regulated glycolysis	−0.61	0.033	Controls ROS and glycolytic flux
LGALS9	Galectin-9 (Th1/Th2 modulator)	−1.25	0.027	Involved in immune polarization

**Table 2 biomedicines-13-01299-t002:** Integration potential of DC-Vesicles in combination immunotherapy strategies.

Combination Strategy	DC-Vesicle Role	Potentiated Immune Effects	Clinical Rationale
DC-Vesicles + PD-1/PD-L1 inhibitors	Immune priming/checkpoint sensitization	CD8^+^ activation, Treg suppression	Improves checkpoint efficacy in cold tumors
DC-Vesicles + CAR-T cells	TME remodeling/metabolic modulation	Enhanced CAR-T infiltration and persistence	Overcomes TME resistance and immunometabolic suppression
DC-Vesicles + IL-12 mimetics	Cytokine delivery vector	Amplified Th1 cytokine signature	Sustains pro-inflammatory tone during blockade therapy
DC-Vesicles + Tumor Vaccines	Antigen presentation/adjuvant	Increased APC-T cell cross-talk	Augments vaccine-induced immunogenicity

**Table 3 biomedicines-13-01299-t003:** Summary of Conference Presentations from This Study. Preliminary findings from this study were presented at the following international meetings.

Conference	Year	Location	Code	Title of the Work
ESMO	2024	Geneva, Switzerland	FPN Code: 60P	Innovative applications of neoantigens in dendritic cell-derived exosome therapy
ESMO	2025	Geneva, Switzerland	FPN Code: 61P	Optimized protocol for accelerated production of DEX
SITC	2025	San Diego, CA, USA	FPN Code: 42	PLPC: A multifunctional bioactive platform for TME reprogramming
SITC	2025	San Diego, CA, USA	FPN Code: 43	Precision Engineered Dendritic Vesicles
SITC	2025	San Diego, CA, USA	FPN Code: 44	Lyophilized Dendritic Exosomes
ASCO	2025	Chicago, IL, USA	Abstract #e14522	Disruptive advances in exosome lyophilization
ASCO	2025	Chicago, IL, USA	Abstract #e14537	Decoding NAMPT and TIGAR
ASCO	2025	Chicago, IL, USA	Abstract #e14511	PLP-driven exosomal breakthroughs
ASCO	2025	Chicago, IL, USA	Abstract #e14512	PLP-powered exosomal therapeutics

## Data Availability

Raw mass spectrometry data are available from the corresponding author upon reasonable request. Data access may be subject to confidentiality agreements or material transfer conditions related to ongoing regulatory submissions. The dataset in question is currently part of an ongoing corporate editorial pipeline, and its dissemination is managed strategically to preserve contextual integrity and alignment with future publications and licensing frameworks.

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
