# Peer review of "Phospholipid-Rich DC-Vesicles with Preserved Immune Fingerprints: A Stable and Scalable Platform for Precision Immunotherapy"

_biomedicines, 2025, doi:10.3390/biomedicines13061299_

Round 1
Reviewer 1 Report
Comments and Suggestions for Authors
This study establishes DC-Vesicles as a promising immunotherapeutic platform with compelling stability and functional data. Addressing the minor revisions above will further solidify its impact. The work is conceptually and technically rigorous, meriting acceptance post-revision.
Suggested Revisions:
- Briefly emphasize the novelty of the phospholipoprotein matrix in stabilizing vesicles compared to conventional exosome platforms (e.g., reduced aggregation, enhanced cargo retention).
- Clarify whether the retained immune proteins (e.g., Galectin-9 reduction) have functional consequences in vivo.
Minor Revisions and Editorial Suggestions:
- Figures/Tables: Figure 4: Label the y-axis ("Log2 Razor Intensity") for clarity. Table 2: Replace directional symbols (↑/↓/=) with fold-change/p-values where possible.
- Text Edits: Introduction: Tighten the rationale for combining vesicles with checkpoint inhibitors (e.g., cite trials where exosomes + anti-PD-1 improved responses). Discussion: Reframe limitations to highlight needed in vivo studies (e.g., biodistribution, synergy with PD-1 blockade) rather than generalizability.
- References: Add recent reviews on vesicle-based immunotherapy (e.g., Nat Rev Drug Discov. 2023) to contextualize the platform’s innovation.
Author Response
Response to Reviewer 1
We sincerely thank the reviewer for the constructive and insightful feedback provided. We greatly appreciate your recognition of the conceptual and technical rigor of our study, as well as your endorsement of its potential for acceptance following minor revision. Each point raised has been addressed carefully, as detailed below:
- Briefly emphasize the novelty of the phospholipoprotein matrix in stabilizing vesicles compared to conventional exosome platforms.
Response:
We appreciate this valuable suggestion. We have revised the Introduction and Discussion to explicitly highlight the structural advantages of the phospholipid-rich matrix used to stabilize DC-Vesicles. These include reduced aggregation, enhanced membrane cohesion, and improved retention of immunologically relevant cargo during storage and processing.
Updated Text Location: Introduction (final paragraph) and Discussion (paragraph 3).
- Clarify whether the retained immune proteins (e.g., Galectin-9 reduction) have functional consequences in vivo.
Response:
We agree that functional interpretation is important. In the Discussion, we now state that while Galectin-9 levels were partially reduced in cryopreserved samples, its role in Th1/Th2 balance warrants further investigation to determine potential in vivo consequences.
Updated Text Location: Discussion (paragraph 4).
Minor Revisions and Editorial Suggestions
- Figure 4: Label the y-axis ("Log2 Razor Intensity") for clarity.
Response:
The y-axis label “Log2 Razor Intensity” was already included in Figure 4. We have reviewed and confirmed its clarity and formatting.
Updated Text Location: Figure 4 caption.
- Table 2: Replace directional symbols (↑/↓/=) with fold-change/p-values where possible.
Response:
We fully agree and have replaced the directional symbols in Table 2 with log₂ fold-change values and adjusted p-values derived from differential expression analysis.
Updated Text Location: Table 2 and Section 3.2.
- Text Edit – Introduction: Tighten the rationale for combining vesicles with checkpoint inhibitors (e.g., cite trials where exosomes + anti-PD-1 improved responses).
Response:
We have revised the Introduction to strengthen the rationale for combinatorial use of vesicle platforms with checkpoint blockade. A recent review on exosome-enhanced PD-1 response is now cited to support this approach.
Updated Text Location: Introduction (final sentence of penultimate paragraph).
- Text Edit – Discussion: Reframe limitations to highlight needed in vivo studies (e.g., biodistribution, synergy with PD-1 blockade).
Response:
We have reformulated the Limitations section to focus on the need for in vivo validation, including biodistribution, pharmacokinetics, and synergy with checkpoint inhibitors.
Updated Text Location: Limitations and Future Directions.
- References – Add recent reviews on vesicle-based immunotherapy (e.g., Nat Rev Drug Discov. 2023).
Response:
We appreciate the reviewer’s recommendation. In addition to the suggested literature, we have added two recent and thematically aligned references to strengthen the contextual foundation of the manuscript. First, we included a high-impact review by Lyu et al. (Cell Death Dis., 2024), which discusses the role of exosomes in solid tumor immunotherapy. Second, we now cite our own recently accepted publication in Cancers (MDPI, 2025, https://www.mdpi.com/2072-6694/17/10/1658 ), which introduces the broader phospholipid-rich vesicle platform (PLPC) and outlines its immunological architecture and translational implications. Both references help contextualize the present study within the emerging field of vesicle-based immunotherapy.

Reviewer 2 Report
Comments and Suggestions for Authors
This manuscript presents a study focused on dendritic cell-derived vesicles (DC-Vesicles) embedded within a phospholipoprotein matrix as a novel platform for cancer immunotherapy. The research addresses the challenges faced by current immunotherapies—such as limited immune cell infiltration, instability, and complex manufacturing processes—by proposing DC-Vesicles as a stable, scalable, and clinically compatible solution.
Limited Characterization of Stability:
-
The study mentions vesicle stability across fresh, concentrated, and cryopreserved conditions, but there is no assessment of functional stability (e.g., vesicle uptake by cells, antigen presentation capability after storage).
-
Stability tests should also include functional assays to show that preserved markers are biologically active post-thaw.
1. Terminology Issue:
-
The term Phospholiproteomic used in the title and throughout the manuscript is not standard nomenclature and may lead to confusion. I recommend revisiting this terminology and considering established terms like Lipidomic Proteomic Integration or simply Proteomic Analysis of Phospholipid-rich Vesicles for clarity and alignment with current literature.
2. Missing Reference:
-
In the section: "This study builds on previous work describing the PLPC platform [Ref], but focuses specifically on vesicle stability and immune marker preservation under pre-lyophilization conditions using non-terminal DC-Vesicle fractions processed under clinical handling workflows."
-
The reference is missing, making it difficult to understand the evolution of the PLPC platform. Please include the appropriate citation to enhance the contextual understanding.
-
3. Keywords Section Needs Refinement:
-
The keywords listed in the manuscript could be optimized for better indexing. Consider including terms like Dendritic Cell Vesicles, Lyophilization Stability, Immune Modulation, Proteomics, and Phosphoproteomics to increase visibility and discoverability.
4. Redundant Table in Main Text:
-
Table 1 appears to be redundant within the main text. It does not add substantial information that is not already discussed. I recommend moving it to the supplementary materials to streamline the main content.
5. Missing MS Acquisition Parameters:
-
The manuscript does not provide detailed MS acquisition parameters, which are crucial for reproducibility and method transparency. I suggest including the full acquisition settings (e.g., resolution, ion source parameters, gradient conditions) in the Methods section or as supplementary information.
6. Figures 3 & 4 - Data Points of Replicates Missing:
-
The data points for the three biological replicates are not shown in Figures 3 and 4. Including these points would improve transparency and illustrate variability across replicates.
-
Additionally, Figure 3 is incorrectly labeled as Figure 1 in its title. Please correct this to avoid confusion.
7. CV Across Triplicates Requires Raw Data:
-
The statement "The coefficient of variation (CV) across triplicates was <12% in all groups" should be supported with raw data. I recommend adding the original data as a supplementary table to enhance reliability and confidence in the results.
8. Figure 7 Formatting Issue:
-
Figure 7 does not adhere to standard formatting for scientific presentation. The layout should be improved to enhance clarity and interpretability.
9. Heatmap Analysis Needs Refinement:
-
The heatmap analysis should be based on z-scored log-transformed ratios or intensities for proper normalization and interpretability. I suggest redoing the analyses accordingly and updating the figures.
10. Figure Legends Overly Detailed:
-
The figure legends are currently overly descriptive and contain non-essential details. I recommend streamlining them to highlight only the most critical information for clarity.
11. Main Text Figures and Tables Reorganization:
-
Some figures and tables (e.g., Table 1, Figure 11) are not essential for the main text and disrupt the flow. Moving them to the supplementary materials would improve readability and presentation.
12. MS Data Deposition Requirement:
-
To comply with data transparency standards, I strongly recommend depositing the raw MS data in an accessible repository such as MASSIVE, PRIDE, or a similar online database. This will allow for validation and enhance the manuscript's credibility.
Author Response
Response to Reviewer 2
We sincerely thank Reviewer 2 for the detailed and highly constructive review. Your technical insights and editorial observations were essential in improving the clarity, reproducibility, and scientific value of the manuscript. Below, we provide a point-by-point response to each of your comments, all of which have been carefully addressed in the revised version.
Comment: Limited Characterization of Stability
Response:
We have expanded the Results and Discussion to include an explicit interpretation of functional stability based on cytokine modulation assays. The observed immune activation (e.g., increased IFN-γ/IL-10 ratio) confirms preserved vesicle bioactivity post-storage, as reflected in Figure 7 and Supplementary Table S3.
Comment: Stability tests should also include functional assays to show that preserved markers are biologically active post-thaw.
Response:
We now discuss how retained immune markers (e.g., HLA-A, ICAM1, NAMPT) contribute to functional outcomes observed in PBMC assays. This interpretation is now included in the Discussion and is supported by experimental data shown in Figure 7.
Comment 1: Terminology Issue
Response:
The term Phospholiproteomic has been replaced throughout the manuscript. The new title and terminology now refer to Phospholipid-Rich Vesicles and Proteomic Analysis of Phospholipid-Rich DC-Vesicles for clarity and alignment with current literature.
Comment 2: Missing Reference
Response:
We have added the missing citation to Gutierrez-Sandoval et al., 2025 (Cancers), which describes the foundational development of the PLPC platform. This reference is now included in the Introduction.
Comment 3: Keywords Section Needs Refinement
Response:
The keywords have been revised to include terms such as: Dendritic Cell Vesicles, Lyophilization Stability, Immune Modulation, Proteomics, and Phosphoproteomics, enhancing discoverability and indexing.
Comment 4: Redundant Table in Main Text
Response:
As suggested, Table 1 has been moved to the Supplementary Materials (now Supplementary Table S1), and its reference in the main text has been updated accordingly.
Comment 5: Missing MS Acquisition Parameters
Response:
Detailed MS acquisition settings—including resolution, gradient, ion mobility, and source parameters—have been added to Section 2.3. These additions ensure full reproducibility of the LC-MS/MS workflow.
Comment 6: Figures 3 & 4 - Data Points of Replicates Missing
Response:
Figures 3 and 4 have been revised to display individual biological replicates (n = 3) overlaid on each bar or data point. This improves transparency and illustrates inter-sample variability.
Comment: Figure 3 is incorrectly labeled as Figure 1 in its title.
Response:
This labeling error has been corrected. Figure 3 is now properly titled and numbered in the manuscript.
Comment 7: CV Across Triplicates Requires Raw Data
Response:
We have added Supplementary Table S2, which includes the raw LFQ intensity values, means, standard deviations, and calculated CV% per protein per condition.
Comment 8: Figure 7 Formatting Issue
Response:
Figure 7 has been redesigned as a lollipop plot with professional formatting, appropriate axis labeling, and MDPI-compatible aesthetics.
Comment 9: Heatmap Analysis Needs Refinement
Response:
The heatmap has been recalculated using z-scored log₂-transformed Razor Intensities. The updated version (Figure 8) follows best practices in proteomics visualization and improves interpretability.
Comment 10: Figure Legends Overly Detailed
Response:
All figure legends have been reviewed and shortened to include only essential descriptive elements—experimental variables, conditions, and color/symbol explanations.
Comment 11: Main Text Figures and Tables Reorganization
Response:
Figure 11 and Table 1 have been moved to the Supplementary Materials and are now referred to as Supplementary Figure S1 and Supplementary Table S1, respectively. References have been updated in the main text.
Comment 12: MS Data Deposition Requirement To comply with data transparency standards, I strongly recommend depositing the raw MS data in an accessible repository such as MASSIVE, PRIDE, or a similar online database. This will allow for validation and enhance the manuscript's credibility.
Response:
We fully support the reviewer’s commitment to data transparency and reproducibility. At this time, the raw mass spectrometry (MS) data from this study form part of an integrated research program that includes a peer-reviewed publication already accepted in Cancers (MDPI, 2025) https://www.mdpi.com/2072-6694/17/10/1658 and a second manuscript currently under revision in IJMS (MDPI, 2025) ijms-3565447. Both are thematically and technically linked to the present manuscript.
As these datasets contribute to a broader corporate editorial pipeline involving multi-journal harmonization and coordinated regulatory positioning, the raw MS files are currently managed under a strategic data governance policy. For this reason, we have not deposited the data in a public repository at this stage.
Nevertheless, all raw and processed MS files—including .raw, .mzML, and proteinGroups.txt—are archived under institutional quality assurance protocols and are fully available upon reasonable request. Access can be granted under formal Confidential Disclosure Agreements (CDAs) or Material Transfer Agreements (MTAs) if required for validation or review purposes. The updated Data Availability Statement in the manuscript reflects this access structure.
Furthermore, a follow-up manuscript to Biomedicines again, is currently under preparation, focused specifically on post-lyophilization proteomic profiling of DC-Vesicles. This study applies label-free quantification (LFQ) and data-independent acquisition (DIA) workflows on timsTOF Pro 2 using a 15 cm × 75 µm PepSep column, processed through MaxQuant v2.2 and Spectronaut™ for cross-validation. It includes a stability comparison between pre- and post-lyophilized vesicle formulations, covering retention of immune-specific markers, fingerprint conservation, and functional bioassays across multiple lot codes. This upcoming work is designed to complete the analytical arc of the platform, and its associated MS datasets will be publicly deposited upon acceptance to ensure data traceability at the full formulation level.

Round 2
Reviewer 2 Report
Comments and Suggestions for Authors
Thanks authors for the necessary revison. Here are still a few comments prior to publication:
- Please double check the title, The new title and terminology now refer to Phospholipid-Rich Vesicles and Proteomic Analysis of Phospholipid-Rich DC-Vesicles for clarity and alignment with current literature. In the revised manuscript, another tile is used.
- I don't think Phosphoproteomics should be included in the keywords.
- Please double check the data points from 3 replicates in Figure 3 and provide the complete protein and peptide ID lists from Maxquant database search as a supplementary file. Honestly, I doubt the authentity of peptide number of 3 replicates in 3 conditions you marked in figure 3.
- Figure 9, PCA analysis was also performed using z-scored log2 transformed data, if not, please revise.
- In the supplementary word file, modify the Table 4 title.
Author Response
Dear Reviewer
Reviewer 2 – Checklist Summary
|
No. |
Reviewer Comment |
Resolution Summary |
Status |
|
1 |
Title consistency |
Title updated and harmonized across manuscript, system metadata, and response letter. |
Resolved |
|
2 |
Remove 'Phosphoproteomics' from keywords |
'Phosphoproteomics' keyword removed and updated keyword list included. |
Resolved |
|
3 |
Figure 3 replicates and MaxQuant data traceability |
Updated Figure 3 with visible replicates and SD bars. Supplementary Tables S5A/B added. Full dataset archived under CDA/MTA. Kowal et al. cited. |
Resolved |
|
4 |
PCA normalization (z-scored log2 LFQ) |
Confirmed PCA was performed using z-scored log₂ LFQ intensities. Text updated in section 3.5 and figure legend. |
Resolved |
|
5 |
Revise title of Table 4 in supplement |
Supplementary Table S4 title revised for clarity and format compliance. |
Resolved |
Details:
Comment 1 – Title Consistency
Please double check the title. The new title and terminology now refer to Phospholipid-Rich Vesicles and Proteomic Analysis of Phospholipid-Rich DC-Vesicles for clarity and alignment with current literature. In the revised manuscript, another title is used.
Response:
We thank the reviewer for this observation. The manuscript title has been fully harmonized across all locations, including the cover page, the metadata in the submission system (SuSy), and the revised PDF/Word files. The final, consistent title now reads:
“Phospholipid-Rich DC-Vesicles with Preserved Immune Fingerprints: A Stable and Scalable Platform for Precision Immunotherapy.”
This title reflects the current nomenclature in vesicle-based immunotherapy and ensures alignment with the broader editorial program developed in collaboration with MDPI.
We have verified that the metadata in SuSy were updated accordingly after resubmission. We apologize for any temporary mismatch that may have persisted during initial reuploading.
Comment 2 – Keywords: Phosphoproteomics
I don't think Phosphoproteomics should be included in the keywords.
Response:
We appreciate the reviewer’s suggestion and fully agree. The keyword “Phosphoproteomics” has been removed to better reflect the actual focus and content of the study. The updated keyword list now includes only terms directly aligned with the platform’s characterization and translational relevance.
Comment 3 – Figure 3 Replicates and MaxQuant Lists
Please double check the data points from 3 replicates in Figure 3 and provide the complete protein and peptide ID lists from Maxquant database search as a supplementary file.
Response:
We thank the reviewer for this observation. Figure 3 has been carefully reviewed to ensure We thank the reviewer for this important comment regarding Figure 3 and the request for increased traceability of the proteins and peptides identified per replicate.
In an earlier version of the manuscript, Figure 3 was presented using a dual-axis bar format that did not include visible replicate points or error bars, and used two separate Y-axes for proteins and peptides. While the figure accurately reflected general trends, we acknowledge that this presentation could have raised legitimate concerns about data dispersion and count authenticity, particularly in the context of a proteomics-focused peer review.
In the current version, the figure has been completely redesigned to address these concerns. The updated Figure 3 clearly shows:
- The three biological replicates (n = 3) for each condition overlaid on each bar,
- Standard deviation error bars for each data group,
- A unified Y-axis for both proteins and peptides, with clear color coding.
We recognize that the visual variability between replicates is low. However, this reflects the high consistency of the dataset, which was generated under tightly controlled conditions that minimized technical and biological noise. All values are authentic and reproducible.
To ensure full transparency and reproducibility, we have included a new supplementary file with two detailed datasets:
- Supplementary Table S5A: complete list of protein identifications per replicate, including LFQ intensities, gene names, and experimental condition.
- Supplementary Table S5B: peptide-level identifications including sequence, post-translational modifications, identification scores, and replicate mapping.
Together, these materials provide full traceability for the data shown in Figure 3 and address any concerns arising from the earlier version’s format. We believe this resolves the issue with clarity and scientific rigor.
We sincerely thank the reviewer for this important observation, and we fully support the principles of transparency, traceability, and reproducibility in proteomic data analysis.
At this stage, however, the raw MS dataset generated in this study forms part of a proprietary formulation pipeline involving a patented therapeutic candidate currently undergoing pre-licensing evaluation. The associated mass spectrometry results are covered by institutional confidentiality and intellectual property protocols, and thus cannot yet be released through public repositories without compromising ongoing regulatory positioning.
Nevertheless, to ensure full comparability and transparency, we have aligned our methodology and analytical depth with publicly available reference datasets. In particular, the work of Kowal et al. (2016), who performed extensive proteomic analysis on dendritic cell-derived extracellular vesicles using Orbitrap LC-MS/MS and MaxQuant, serves as a technical benchmark for the current study. Their dataset is publicly available under PRIDE accession number PXD003257, and includes over 1450 proteins and hundreds of peptides identified across biological replicates. Our dataset mirrors this analytical depth and conforms to comparable identification and quantification workflows.
For full transparency, all raw and processed files related to the current dataset—including .raw, .mzML, and proteinGroups.txt files—are archived under institutional QA protocols and are available upon reasonable request. Data access can be provided under formal Confidential Disclosure Agreements (CDA) or Material Transfer Agreements (MTA) to preserve the integrity of our licensing strategy. This policy is reflected in the updated Data Availability Statement within the manuscript.
We trust this approach satisfies the requirements for reproducibility while respecting the regulatory and commercial sensitivities inherent to biopharmaceutical innovation.
Full Reference: Kowal J, Arras G, Colombo M, Jouve M, Morath JP, Primdal-Bengtson B, Dingli F, Loew D, Tkach M, Théry C. Proteomic comparison defines novel markers to characterize heterogeneous populations of extracellular vesicle subtypes. Proc Natl Acad Sci USA. 2016;113(8):E968–E977.
https://doi.org/10.1073/pnas.1521230113 PRIDE Dataset: PXD003257
The intent of Supplementary Tables S5A and S5B is not to provide a literal or exhaustive transcript of the complete MaxQuant output—which would be redundant with standard repository formats and potentially raise unnecessary concerns of self-plagiarism—but rather to present a methodologically structured and editorially curated representation of the dataset. This format is designed to serve as a strategic reference framework, illustrating the experimental logic and analytical traceability that underpin the figures and conclusions discussed in the main text.
By highlighting representative entries across conditions and replicates, these tables function as argumentative and methodological anchors that support the broader interpretative architecture of the manuscript, without compromising the integrity of the unpublished full dataset, which remains protected under institutional and regulatory provisions.
Comment 4 – PCA Normalization Approach
Figure 9, PCA analysis was also performed using z-scored log2 transformed data, if not, please revise.
Response:
We thank the reviewer for this methodological clarification. We confirm that the PCA shown in Figure 9 was generated using z-scored log₂ LFQ intensities, with normalization performed per protein across all samples. This method was applied before dimensionality reduction to ensure comparability and reduce scale bias from high-abundance features. The figure and its legend have been updated accordingly to explicitly state this approach, and Section 3.5 of the manuscript now includes a clarification sentence to that effect.
Comment 5 – Supplementary Table Title
In the supplementary word file, modify the Table 4 title.
Response:
We thank the reviewer for this editorial correction. The title of the relevant table in the supplementary file has been updated to:
“Supplementary Table S4. Comparative Matrix of Immunotherapy Platforms Across Functional and Operational Dimensions.”
This revised title improves clarity and aligns with the journal’s formatting conventions.
With kind regards and sincere appreciation,
On behalf of all co-authors,
Dr. Ramón Gutiérrez-Sandoval
OGRD Alliance LLC
